# Willingness to participate in, support or carry out scientific studies for benefit assessment of available medical interventions: A stakeholder survey

Julia Stadelmaier[1¤], Joerg J. Meerpohl[1,2¤], Ingrid Toews[1¤]*

**1** Institute for Evidence in Medicine, Medical Centre - University of Freiburg, Faculty of Medicine, University of Freiburg, Freiburg, Germany, **2** Cochrane Germany, Cochrane Germany Foundation, Freiburg, Germany

¤ Current address: Department of Data Driven Medicine, Institute for Evidence in Medicine, Medical Centre - University of Freiburg, Faculty of Medicine, University of Freiburg, Freiburg, Germany
* toews@ifem.uni-freiburg.de

## Abstract

**Data Availability Statement:** All relevant data are within the manuscript and its Supporting information files.

### Background

Post-entry studies are a key element in managed entry agreements and aim at generating evidence about the additional benefit of new medical interventions before reimbursement decisions are made. This study evaluates the willingness of different stakeholder groups to engage post-entry in studies for benefit assessment and to assess differences in their willingness by study type, i.e. randomised controlled trial or observational study.

### Methods

We conducted a cross-sectional, web-based survey with a self-administrated questionnaire in German language. We disseminated invitations to patients, patient representatives, healthcare providers, trialists & scientists and representatives of the medical private sector, using a snowball system, public contact details of German associations and organisations, and social media. We analysed quantitative data descriptively and qualitative data inductively.

### Results

Data of 154 respondents were available for analysis. The majority (>85%) was willing to engage in the studies in general, and regarding different study types. Scientists reported a higher willingness to conduct and support RCTs (p = 0.01) as compared to observational studies. Representatives of the private sector were mainly willing to support, but not to carry out post-entry studies. Stakeholders frequently mentioned that potential personal benefit and altruistic motives were relevant for their decision to engage in studies. Practical inconveniences, poor integration into daily life, high demand for time and personnel, and lack of resources were commonly mentioned barriers.

**Funding:** The study is part of an overarching project on the value of RCTs and observational studies for benefit assessment funded by the National Association of Statutory Health Insurance Funds Germany (SV 19-16). The funder had no role in collecting and interpreting the data. The article processing charge was funded by the Baden-Württemberg Ministry of Science, Research and Art and the University of Freiburg in the funding programme Open Access Publishing.

**Competing interests:** The authors have declared that no competing interests exist.

## Discussion and conclusion

Stakeholders clearly reported to be willing to engage in post-entry studies for benefit assessment. Self-reported willingness to participate in and support for studies seems higher than practical recruitment rates. The survey might be subject to survey error and self-enhancement of participants. Inquiring about the willingness of hypothetical studies might have caused participants to report higher willingness. Motives for and against participation may be possible starting points for approaches to overcome recruitment difficulties and facilitate successful study conduct.

## Introduction

Healthcare decision-makers have the mandate to fairly allocate limited resources while ensuring a high-quality healthcare service provision to achieve the best possible health outcomes of the population [1]. Reimbursement decisions are instruments to facilitate or impede the access to healthcare interventions for target populations. Since healthcare interventions are constantly being developed and improved, countries have often established a system of assessing new interventions for their additional benefit before reimbursement decisions are made. Assessment is often done in health technology assessments where new interventions are examined with regard to benefits, harms, safety, efficacy and cost-effectiveness, considering the currently best available evidence [2].

Generally, a more timely and wider access to healthcare innovations is desired, so that health service users can benefit more rapidly from medical innovations [3]. However, reimbursement decision-making without sufficiently informative evidence is not sensible, and accelerating the availability of innovations often conflicts with requirements of quality and safety, and cost-effectiveness imperatives for healthcare [4]. The implementation of managed entry agreements (MEA) represents one approach to address this conflict. In MEA, new interventions are made available while, in parallel, more evidence on intervention effects is generated. New innovative interventions are reimbursed temporarily or until sufficient evidence is available for final decision-making ("post-entry studies") [5]. In Germany, for example, this approach is used as per §137e Social Code Book V and offers temporary, conditional reimbursement for interventions with the "*potential of a necessary treatment alternative*" [6].

Post-entry studies are a key element in this approach and generate further evidence about intervention effectiveness for decision-making. In light of the particularity of post-entry studies, i.e. they have to be carried out while the intervention is already used in practice, it is important to reflect on their acceptability and feasibility. Since recent regulation by the German Federal Ministry of Health aims to include evidence of different sources [7], both, randomised controlled trials (RCTs) and observational studies (OSs) need to be considered as applicable study types for post-entry studies for benefit assessment.

RCTs are considered the "gold standard" for assessing the impact of interventions. Therefore, RCTs and systematic reviews of RCTs are preferred source of evidence for healthcare policy and decision-making on individual, clinical, and regulatory level. However, conducting RCTs can be challenging due to legal, social or ethical reasons. In addition, RCTs entail a high administrative burden [8]. On the other hand, observational studies are considered to be less challenging; however, they have limited control over confounding factors. Nonetheless, they are often utilised for research questions where conducting RCTs is challenging [8].

> ## Box 1. Working definitions for this study.
>
> *Willingness to participate*: readiness of health service users to participate as volunteers in a study
>
> *Willingness to support*: readiness to engage in a study in a supportive role / function (e.g. assistance in the recruitment, execution of the intervention); private sector: readiness to subsidise a study with financial or in kind resources.
>
> *Willingness to carry out*: readiness to engage in a study as study sponsor (leading role in and responsibility for the planning, execution and evaluation of the study).

Stakeholders' willingness to contribute to research studies is an important determinant of the feasibility of (post-entry) benefit assessment studies. Stakeholders' roles can be to participate in studies, support studies, or carry out studies (see Box 1). The acceptance of the study by all stakeholders and their engagement influence whether studies can be carried out successfully [9, 10]. Previous research examined the willingness of stakeholders to engage in research in general–especially with regard to participants [11], or the willingness regarding a specific study type [12]. However, there seems to be no evidence that deals with the willingness, particularly in post-entry studies for benefit assessment. So, this study aims to evaluate the willingness of relevant stakeholder groups to participate or engage in (support or carry out) post-entry studies. Furthermore, it evaluates relevant motives and concerns of stakeholders and draws a comparison between different study types.

## Materials and methods

### Study design

We conducted a cross-sectional, web-based, open survey aimed at patients (health service users), patient representatives, scientists, healthcare professionals and representatives of the private sector, that are involved in studies of interventions that are already widely available. We reported this study in accordance with the Checklist for Reporting Results of Internet E-Survey (CHERRIES) [13]. Ethical approval was obtained from the Ethics Committee of the Albert-Ludwigs-University Freiburg (No.294/20).

### Questionnaire

In the course of the systematic literature searches we did not identify any suitable, validated questionnaire for the specific purpose of our study. Therefore, we designed a de-novo self-administered questionnaire in German language. Answer options were developed on the basis of a literature research and three expert consultations. Questions with single- or multiple-choice and free-text fields were used. Willingness to participate in RCTs and OSs, respectively, was measured with visual analogue scales ranging from 0 to 100 with higher values indicating higher willingness.

The 51-item questionnaire was sub-divided into five sections: (i) identification of participants as stakeholder, (ii) general willingness to participate, support or carry out studies, (iii) willingness to participate, support or carry out RCTs or observational studies, respectively, (iv) demographic and professional background including experience in years (for patient representatives and representatives of the private sector) or number of engagements in studies of

medical research (for patients, healthcare providers and scientists), and (v) additional comments. With the help of a survey-logic, the identification as a specific stakeholder determined which items of the questionnaire were displayed in the subsequent pages. Items, vocabulary and expressions were selected carefully in order to prevent linguistic miscomprehension which could lead to false responses. In multiple-choice questions, answer options were displayed in a random order to reduce bias due to the primacy effect [14]. The selection was restricted to three answer options in order to identify the most relevant motives. The questionnaire with all items is attached in the S1 Appendix.

We did not perform psychometric testing to validate our questionnaire. However, we carried out a pre-test of the questionnaire with 12 members of our working group and experts in health research methods. The pre-test aimed to identify potential issues in the functioning and wording of the survey, and to finalise all answer options.

## Survey administration

The questionnaire was created on the online platform SoSci Survey [15] and it was open for participation for five weeks, from 15th June to 19th July 2020. A reminder was sent two and four weeks after initiation of the survey. Participation in the study was voluntary, and respondents could terminate their participation in the survey at any time and without giving reasons. The survey ended when the respondent closed their browser window. No reward was offered for participation, but the opportunity to be informed about the results of the study was provided. No strategy to avoid multiple entries was implemented in order to guarantee the anonymity of the participants. Study information, information about data protection, and the contact information of the investigator were presented in the survey introduction. No personal but only anonymised data were collected and analysed for the purpose of this study. Therefore, no informed consent for the study was necessary according to the standards of the Ethics Committee of the Albert-Ludwigs-University Freiburg and the data protection officer. Respondents automatically participated by progressing to the first page of the survey.

None of the questions of the survey were mandatory, and all except for the question used to identify the participants' stakeholder group(s) could be skipped. In the case of non-selection of any answer option, a note appeared to encourage the participants to answer. A button to return to the previous page was displayed to allow respondents to review and change previous answers.

## Sampling

Members of the following stakeholder groups were the target population of the survey: (i) patients or patient representatives, (ii) healthcare providers (physicians and health professionals), (iii) scientists and trialists in medical and health science, and (iv) representatives of the medical or pharmaceutical private sector (e.g. medical technology or the pharmaceutical industry). No geographical limitation was applied. However, as the questionnaire was written in German, knowledge of the German language was necessary to participate in the survey.

We used a snowball system for recruitment. Representatives of German associations and societies of the stakeholder groups were invited to take part in the survey and to forward the invitation to members of their network. A google search was performed to identify relevant associations and societies representing the target stakeholder groups in Germany. In order to recruit patients and patient representatives, the office of patient representatives in the German Federal Joint Committee (Gemeinsamer Bundesausschuss, G-BA), as well as the four organisations that are currently entitled to appoint patient representatives to the G-BA were contacted. Healthcare providers (physicians, dentists, psychologists, nurses, allied health professionals)

were contacted via associations representing the interests of their respective profession. Scientists and trialists were recruited via representatives of the German network for evidence-based medicine, the German network for healthcare research and the Clinical Trials Unit of the University of Freiburg. Trade organisation for the medical technology and pharmaceutical industries were approached in order to invite representatives of the private sector. The contacted associations and societies are listed in detail in the S2 Appendix.

We disseminated email invitations to the target organisations via public contact details. The invitation included a brief introduction to the study, an approximate completion time and the link to the survey. In addition, an invitation to the survey was disseminated via Facebook and Twitter accounts of Cochrane Germany and their newsletter [16].

### Data management

Data were collected, documented and structured on the basis of an a priori developed study protocol. Data were managed by the research team. All data were collected anonymously. The collected data were coded for statistical analysis and will be stored for five years in the password protected server environment of the Albert-Ludwigs University, Freiburg.

### Data analysis

Data were downloaded from SoSci Survey once the survey was closed. Participants who provided at least two valid responses were included in the analysis. However, respondents who did not identify themselves as belonging to one of the stakeholder groups and who did not answer at least one additional question were excluded from the analysis. All valid responses were tabulated, and statistical analyses were conducted using Microsoft Excel 2010. Analyses were performed for each stakeholder group individually. The participation rate was evaluated by dividing the number of people who had completed at least one survey page by the total number of visitors to the survey. The completion rate was calculated using the ratio of people who had submitted the final questionnaire page and those who had completed the first question page.

Motives were described according to their frequency and percentage, and were ranked accordingly. Answers to the visual rating scales were divided into five groups and recoded as follows: "1 to 20" as very low, "21 to 40" as low, "41 to 60" as moderate, "61 to 80" as high, and "81 to 100" as very high. For comparison between two groups an independent t-test was used for a response frequency $n > 30$, and the Mann-Whitney-U-Test for $n < 30$. Results with $p < 0.05$ were considered to be statistically significant. Qualitative data from the free-text responses were analysed inductively by coding the data and creating themes.

## Results

Of the 606 visitors of the survey, 185 edited the questionnaire, resulting in a participation rate of 30.5%. A total of 139 respondents submitted the final questionnaire page, so the completion rate was 75.1%. Seven participants were excluded because they did not identify themselves as belonging to any stakeholder group, and 24 questionnaires were excluded due to missing responses. Hence, data from a total of 154 participants were included in the analysis.

### Demographic characteristics of the sample

Data of 46 patients and patient representatives, 72 healthcare providers, 48 scientists and 13 representatives of the private sector were included in the analysis; 24 of the participants

**Table 1. Characteristics of the sample.**

| | Patient | | Patient representatives | | Healthcare provider | | Scientist | | Private sector | | Total[a] | |
|---|---|---|---|---|---|---|---|---|---|---|---|---|
| | N | % | N | % | N | % | N | % | N | % | N | % |
| **Gender** | | | | | | | | | | | | |
| Female | 26 | 56.5 | 8 | 47.0 | 35 | 48.6 | 30 | 62.5 | 3 | 23.1 | 81 | 52.6 |
| Male | 12 | 26.1 | 7 | 41.2 | 26 | 36.1 | 12 | 25.0 | 7 | 53.8 | 48 | 31.2 |
| Divers | 0 | 0.0 | 0 | 0.0 | 1 | 1.4 | 1 | 2.1 | 1 | 7.7 | 2 | 1.3 |
| Not specified | 8 | 17.4 | 2 | 11.8 | 10 | 13.9 | 5 | 10.4 | 2 | 15.4 | 23 | 14.9 |
| Total | 46 | 100.0 | 17 | 100.0 | 72 | 100.0 | 48 | 100.0 | 13 | 100.0 | 154 | 100.0 |
| **Age** | | | | | | | | | | | | |
| 18 to 24 years | 1 | 2.2 | 0 | 0.0 | 1 | 1.4 | 0 | 0.0 | 3 | 23.1 | 3 | 1.9 |
| 25 to 34 years | 7 | 15.2 | 0 | 0.0 | 12 | 16.7 | 9 | 18.8 | 7 | 53.8 | 21 | 13.6 |
| 35 to 44 years | 7 | 15.2 | 0 | 0.0 | 20 | 27.8 | 18 | 37.5 | 1 | 7.7 | 40 | 26.0 |
| 45 to 54 years | 9 | 19.6 | 3 | 17.6 | 15 | 20.8 | 10 | 20.8 | 0 | 0.0 | 32 | 20.8 |
| 55 to 64 years | 8 | 17.4 | 6 | 35.3 | 14 | 19.4 | 4 | 8.3 | 0 | 0.0 | 26 | 16.9 |
| 65 years or older | 6 | 13.0 | 6 | 35.3 | 1 | 1.4 | 2 | 4.2 | 0 | 0.0 | 8 | 5.2 |
| Not specified | 8 | 17.4 | 2 | 11.8 | 9 | 12.5 | 5 | 10.4 | 2 | 15.4 | 24 | 15.6 |
| Total | 46 | 100.0 | 17 | 100.0 | 72 | 100.0 | 48 | 100.0 | 13 | 100.0 | 154 | 100.0 |

[a] Totals are unequal to the sum of respondents in each stakeholder group since n = 24 of the respondents identified themselves as belonging to two or three stakeholder groups

identified themselves as belonging to two or three stakeholder groups. The characteristics of each stakeholder group are presented in Tables 1–3.

The majority of respondents were female (52.6%), aged between 35 and 54 years (46.8%), and reported to have little experience with regard to participation or engagement in medical research.

## General willingness to engage

The majority of the respondents reported to be willing, in principle, to engage in studies of benefit assessment of interventions that were already used in practice. Forty-two patients (91.3%) were willing to participate; all patient representatives (n = 17) and 68 of the healthcare providers (95.8%) were willing to support, and 40 of the scientists (87.9%) were willing to carry out studies for benefit assessment post-entry. Ten out of thirteen respondents of the private sector (76.9%) were also willing to support, but only four of them were willing to carry out such studies.

**Table 2. Engagement of the respondents in medical research (in number of studies)[a].**

| | Patient | | Healthcare provider | | Scientist | |
|---|---|---|---|---|---|---|
| | N | % | N | % | N | % |
| None | 16 | 34.8 | 20 | 27.8 | 3 | 6.3 |
| 1 to 5 studies | 21 | 45.7 | 29 | 40.3 | 24 | 50.0 |
| 6 to 10 studies | 1 | 2.2 | 5 | 6.9 | 3 | 6.3 |
| 11 to 15 studies | 0 | 0.0 | 3 | 4.2 | 1 | 2.1 |
| > 15 studies | 0 | 0.0 | 4 | 5.6 | 1 | 2.1 |
| Not specified | 8 | 17.4 | 11 | 15.3 | 16 | 33.3 |
| Total | 46 | 100.0 | 72 | 100.0 | 48 | 100.0 |

**Table 3. Experience of the respondents in medical research (in years)[a].**

| | Patient representatives | | Private sector | |
|---|---|---|---|---|
| | N | % | N | % |
| <1 year | 0 | 0.0 | 0 | 0.0 |
| 1 to 5 years | 4 | 23.5 | 2 | 15.4 |
| 6 to 10 years | 7 | 41.2 | 4 | 30.8 |
| >10 years | 4 | 23.5 | 6 | 46.2 |
| Not specified | 2 | 11.8 | 1 | 7.7 |
| Total | 17 | 100.0 | 13 | 100.0 |

[a] Wording and scales of the item 'experience' differed between the respective stakeholder groups.

When asked for their motives for engagement, personal interest and altruistic motives such as the improvement of healthcare or interventions dominated. Burden associated with the engagement and lack of resources and finance were commonly mentioned as reasons against engagement. However, eleven (24.4%) patients, eight (47.1%) patient representatives, eleven (15.5%) healthcare providers and four (9.8%) scientists indicated that they did not see any reasons against participating, supporting or carryout. The five most frequently mentioned motives in each stakeholder group are displayed in Table 4. All motives with their frequency of being mentioned are listed in S3 Appendix.

## Willingness with regard to study type

Fig 1 shows the self-reported willingness to engage in post-entry studies in each stakeholder group in RCTs and OSs, respectively. Except for the private sector, the majority (at least 60%) of the respondents reported high or very high willingness to participate, support or carry out both study types. The mean score for willingness to participate in RCTs was 65.5 points (SD 28.5) as compared to 75.7 points (SD 25.1) in OSs. The difference in willingness to participate in RCTs or OSs was not significant (p = 0.08). Regarding willingness to support RCTs by patient representatives the mean was 78.7 points (SD 26.3) With regard to supporting OSs, the mean was 73.2 points (SD 26.5). No significant difference in willingness to support RCTs or OSs was revealed (p = 0.08). Regarding willingness to support by healthcare providers, the mean was 78.7 points (SD 26.3) and 73.2 points (SD 26.5) for RCTs and OSs, respectively. The difference in the willingness to support between RCTs and OSs was not significant (p = 0.24). The mean score for willingness in respondents of the private sector to support was 42.0 points (SD 26.9) for RCTs and 60.8 points (SD 30.0) for OSs. The difference was not significant (p = 0.23). The mean of willingness of respondent from the private sector to carry out a study was 26.6 points (SD 34.8) for RCTs and 33.3 points (SD 30.8) for OSs. The difference was not significant (p = 0.42). The mean of willingness of scientists to carry out a study was 85.3 points (SD 22.8) and 71.0 points (SD 28.6) for RCTs and OSs, respectively. The difference between RCTs and OSs was statistically significant (p = 0.01).

The most frequently selected motives which referred to specific study types are displayed in Tables 5 and 6. All motives with their frequency of being mentioned are listed in S4 Appendix.

## Discussion

### Summary of the key results

Data of 46 patients, 72 healthcare providers, 48 scientists and 13 representatives of the private sector were analysed. In general, the majority of patients, patient representatives, healthcare

**Table 4. Principal motives for or against engaging in post-entry studies (Top 5).**

| | | Motives for engagement | | | Motives against engagement | | |
|---|---|---|---|---|---|---|---|
| **Participation** | **Patients** | N | % | | | N | % |
| | Personal relevance | 22 | 52.4 | Poor integration into daily life | | 23 | 51.1 |
| | Improvement of quality of methods | 22 | 52.4 | Strain and negative impact | | 17 | 37.8 |
| | Improvement of healthcare | 20 | 47.6 | Lack of trust in study personnel | | 15 | 33.3 |
| | Benefits for future patients | 19 | 45.2 | Lack of personal relevance | | 7 | 15.6 |
| | Interest | 16 | 38.1 | Lack of necessity | | 6 | 13.3 |
| **Support** | **Patient representatives** | | | | | | |
| | Improvement of healthcare | 10 | 58.8 | Strain for participants | | 6 | 35.3 |
| | Interest | 8 | 47.1 | Lack of trust in study personnel | | 4 | 23.5 |
| | Improvement of quality of methods | 8 | 47.1 | Lack of necessity | | 4 | 23.5 |
| | Benefits for future patients | 8 | 47.6 | Poor integration into daily life | | 4 | 23.5 |
| | Trust in study personnel | 7 | 41.2 | Lack of clinical relevance | | 4 | 23.5 |
| | **Healthcare providers** | | | | | | |
| | Improvement of healthcare | 42 | 59.2 | Poor integration into daily life | | 33 | 46.5 |
| | Benefits for future patients | 36 | 50.7 | Lack of resources | | 27 | 38.0 |
| | Improvement of quality of methods | 29 | 40.8 | Lack of personal relevance | | 19 | 26.8 |
| | Personal relevance | 25 | 35.2 | Strain for participants | | 18 | 25.4 |
| | Interest | 24 | 33.8 | Insufficient compensation | | 14 | 19.7 |
| | **Private sector** | | | | | | |
| | Progress of own methods | 6 | 60.0 | Additional costs | | 10 | 76.9 |
| | Improvement of quality of methods | 5 | 50.0 | Already sufficient evaluation | | 6 | 46.2 |
| | Improvement of healthcare | 4 | 40.0 | Lack of relevance | | 5 | 38.5 |
| | Marketing for method | 4 | 40.0 | Lack of necessity | | 2 | 15.4 |
| | Provision of additional data | 3 | 30.0 | Lack of benefits | | 2 | 15.4 |
| | Relevance | 3 | 30.0 | | | | |
| **Carryout** | **Private sector** | | | | | | |
| | Improvement of quality of methods | 2 | 50.0 | Additional costs | | 10 | 83.3 |
| | Relevance | 2 | 50.0 | Already sufficient evaluation | | 8 | 66.7 |
| | Benefits for future patients | 2 | 50.0 | Finance | | 5 | 66.7 |
| | | | | Difficulties in implementation | | 2 | 25.0 |
| | | | | Lack of necessity | | 2 | 16.7 |
| | **Scientists** | | | | | | |
| | Improvement of quality of methods | 22 | 47.8 | Insufficient research funds | | 18 | 43.9 |
| | Improvement of healthcare | 22 | 47.8 | Lack of relevance | | 13 | 31.7 |
| | Benefits for future patients | 18 | 39.1 | Lack of resources | | 12 | 29.3 |
| | Interest | 17 | 37.0 | Limitations in methods | | 8 | 19.5 |
| | Personal relevance | 16 | 34.8 | Problems in implementation | | 8 | 19.5 |

providers and scientists were willing to participate or engage in studies, also with regard to a specific study type (RCTs and OSs). The most important motives for or against participation or engagement were similar across the four stakeholder groups. Personal benefit and interest, as well as benefits for others including the desire to contribute to the improvement of interventions and healthcare in general were most frequently mentioned. In contrast, poor integration into daily life of tasks associated with the studies, expectations of strain and negative impacts, and lack of trust in study personnel were mentioned as reasons against participation or engagement. At least 60% of all individual stakeholder groups showed high to very high levels of acceptance for both RCTs and OSs.

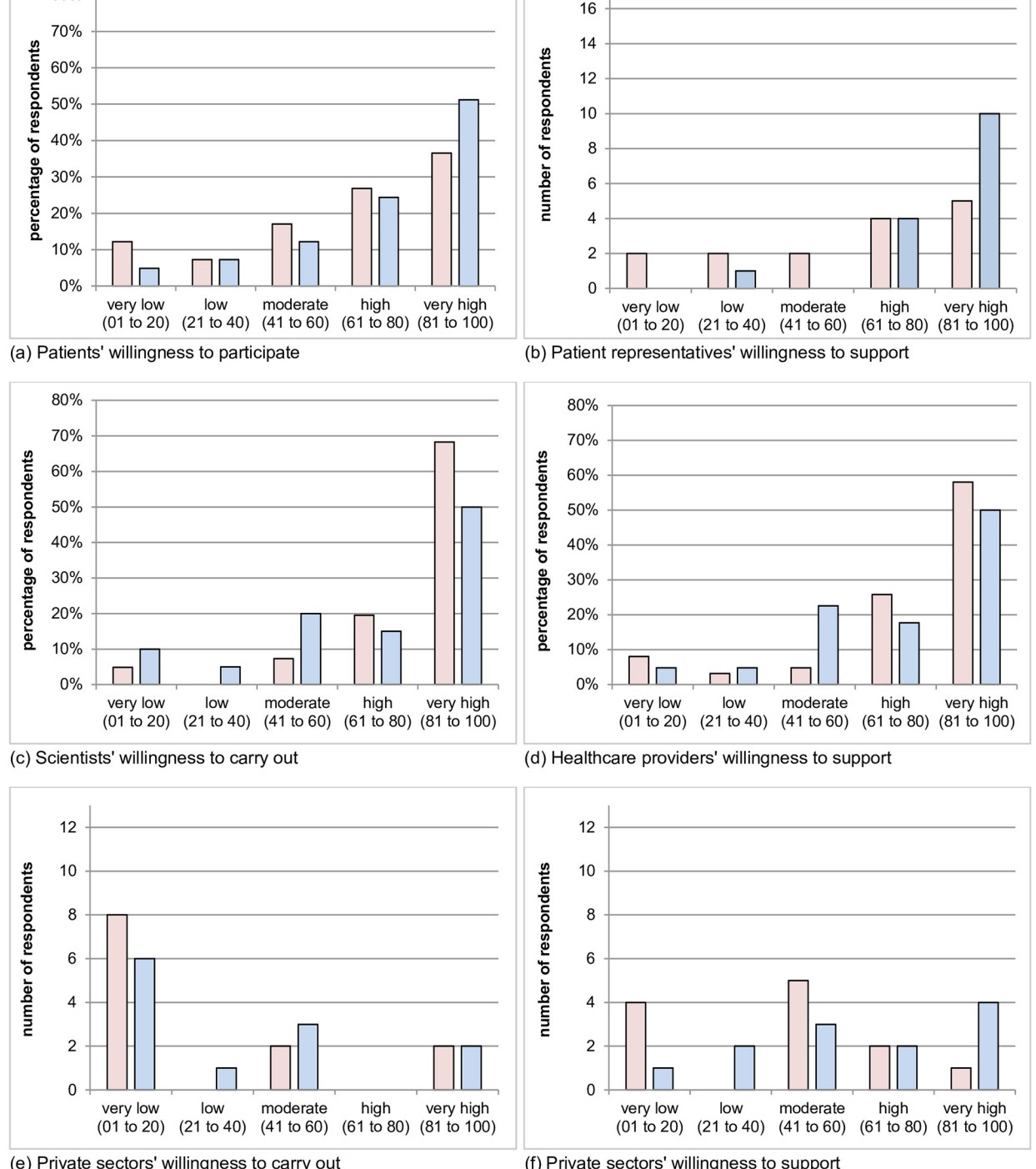

**Fig 1. Willingness with regard to study type.** Red: randomised controlled trials; blue: observational studies.

Representatives of the private sector varied in their willingness to support RCTs and OSs. For both study types, they were motivated by a wish to progress their own methods, to facilitate marketing, and to contribute to the improvement of the quality of methods and healthcare. However, they were concerned about additional costs and source of finance, and

**Table 5. Motives against engagement in randomised controlled trials.**

| Participation | Patients | N | % |
|---|---|---|---|
| | Probable withholding of a method | 17 | 41.5 |
| | *No relevant reasons* | 16 | 39.0 |
| | No control over intervention allocation due to randomisation | 12 | 29.3 |
| | Worse health outcomes expected due to different treatments | 9 | 22.0 |
| Support | **Patient representatives** | | |
| | Withholding a method from the control group | 6 | 40.0 |
| | *No relevant reasons* | 5 | 33.3 |
| | No control over intervention allocation due to randomisation | 4 | 26.7 |
| | Worse health outcomes expected due to different treatments | 4 | 26.7 |
| | **Healthcare providers** | | |
| | *No relevant reasons* | 29 | 46.8 |
| | Withholding a method from the control group | 16 | 25.8 |
| | Worse health outcomes expected due to different treatments | 12 | 19.4 |
| | Probable disagreement with assigned method | 10 | 16.1 |
| | **Private sector** | | |
| | Withholding own method from the control group | 6 | 60.0 |
| | Lack of relevance | 5 | 50.0 |
| | Lack of interest | 4 | 40.0 |
| | No influence on comparator | 3 | 40.0 |
| Carryout | **Private sector** | | |
| | High demand for time and personnel | 11 | 91.7 |
| | Lack of relevance | 9 | 75.0 |
| | Lack of interest | 4 | 33.3 |
| | High risk of discontinuation of the study expected due to lack of healthcare providers' willingness to engage | 3 | 25.0 |
| | **Scientists** | | |
| | *No relevant reasons* | 23 | 56.1 |
| | High demand for time and personnel | 7 | 17.1 |
| | Expected difficulties in recruitment of volunteers | 7 | 17.1 |
| | High risk of discontinuation of the study expected due to lack of patients' willingness to participate | 7 | 17.1 |
| | High risk of discontinuation of the study expected due to lack of healthcare providers' willingness to engage | 7 | 17.1 |

considered that there was no necessity for the studies since there had already been sufficient evaluation pre-entry.

## Interpretation

Based on the high self-reported willingness to participate in, support or carry out post-entry studies, positive engagement can be assumed on the part of patients, patient representatives, healthcare providers and scientists when planning and conducting post-entry studies for benefit assessment. This finding is conflicting with findings in the scientific literature that participant recruitment and adherence to studies is challenging in many instances, and a substantial proportion of studies fail due to difficulties in recruitment [17, 18]. The result of this study could indicate that the recruitment of patients or healthcare providers might be easier if the intervention is already available. Due to wide availability, potential participants might consider interventions as safe, and therefore may be more inclined to participate or promote

**Table 6. Motives against engaging in observational studies.**

| Participation | Patients | N | % |
|---|---|---|---|
| | *No relevant reasons* | 24 | 58.4 |
| | Lack of benefit due to absence of a control group | 10 | 24.4 |
| | High burden due to additional examinations | 8 | 19.5 |
| Support | **Patient representatives** | | |
| | *No relevant reasons* | 11 | 73.3 |
| | Lack of benefit for participants expected | 2 | 13.3 |
| | Others | 2 | 13.3 |
| | **Healthcare providers** | | |
| | *No relevant reasons* | 29 | 46.8 |
| | Low certainty of findings of OSs | 25 | 40.3 |
| | Lack of benefit due to absence of control group | 16 | 25.8 |
| | No benefit for participants expected | 7 | 11.3 |
| | **Private sector** | | |
| | Lack of benefit due to absence of control group | 4 | 33.3 |
| | Confounding due to low adherence high drop-out | 4 | 33.3 |
| | Lack of interest | 3 | 25.0 |
| Carryout | **Private sector** | | |
| | High demand for time and personnel | 7 | 58.3 |
| | Lack of benefit due to absence of control group | 4 | 33.3 |
| | Lack of interest | 4 | 33.3 |
| | **Scientists** | | |
| | Low certainty in findings of OSs | 19 | 46.3 |
| | *No relevant reasons* | 16 | 39.0 |
| | Lack of benefit due to absence of control group | 14 | 34.1 |
| | High demand for time and personnel | 5 | 12.2 |

We did not carry out further subgroup analyses on the effect of sociodemographic variables since the number of respondents in each stakeholder group is too small to create further representative subgroups and perform reasonable analyses.

participation in studies [19]. However, it should be noted that only general questions about willingness were asked and no specific scenario was presented. Thus, inquiring about the willingness of hypothetical studies might have caused respondents to report higher willingness. Furthermore, respondents in the group of healthcare providers lack individual experience in medical research and might not be aware of these challenges in practice.

Knowledge about stakeholders' motives for and against participation and engagement is important in practice since it can serve as starting point for overcoming challenges or barriers in planning and implementing post-entry studies. Personal benefit and altruistic motives were key themes in all stakeholder groups that are also present in current literature [11, 20]. Information on the relevance, necessity and potential benefits of a specific study might positively affect potential volunteers' decision to participate, and potential partners' (healthcare providers, patient representatives) decision to engage. Therefore, in the run-up to any study, emphasis should be put on the provision of information and education of stakeholders. This strategy is especially important in post-entry studies, since, as in this survey, health service users or providers might not see direct benefits for themselves or their patients, respectively, or benefits for future health service users since interventions are already wide available.

With regard to barriers to participation or engagement, many participants identifying themselves as patients or healthcare providers mentioned poor integration into daily life of study tasks and high levels of inconvenience as common reasons for concerns. This is in accordance with findings in existing scientific literature that point to additional visits or distances to trial or examination centres as relevant factors for participation [12, 19, 21]. In order to facilitate the implementation of studies, researchers might need to design easily feasible ways to integrate study tasks into daily life—for health service users, but also for recruiting or examining parties.

Stakeholders' perceptions of study type characteristics such as lack of control of intervention allocation due to randomisation is an important source of uncertainty regarding participation in RCTs, as it is also highlighted in a recent Cochrane review [19]. This may be due to a lack of understanding of the concept of randomisation and the principle of clinical equipoise between different study arms. When potential study participants assume clear differences in health outcomes between intervention and control group, for example, respondents' concerns of not receiving the best intervention available and having poorer outcomes in the control group are comprehensible. Furthermore, treatment preferences are a key factor in the decision to participate in trials [19], but these preferences cannot be taken into account when group assignment is random. However, since we did not provide details of a specific study in this survey, it remains unclear which type of interventions and control interventions the respondents pictured when answering the question, and to which comparison the concerns are linked. Future research that specifies details of the comparator (active or inactive) might therefore be indicated. This could contribute to resolving the paradox between high willingness of study participation as reported in our survey and other research findings that demonstrate a reluctance of study participation in general [19].

Only few representatives of the private sector took the survey. It remains unclear whether the invitation was not forwarded by intermediaries of pharmaceutical organisations/ associations, for example due to a lack of relevance or permission, or whether the topic was generally seen as not relevant by this stakeholder group. However, willingness seemed to be linked to the aspects of financing; and respondents appeared to be unwilling to conduct a study if funding had to be provided entirely by their company.

## Strengths and limitations

Strengths of the survey include our rigorous method of implementation. The survey was conducted online, which allowed participants to respond at their own pace. Honest responses were encouraged by the online-based conduct of the survey. Each participant was able to conduct the survey in a private setting, removing themselves from the possible influence or judgment of third parties regarding the responses. Another strength of the study was an appropriate number of participants from three of four stakeholder groups.

This study also presents several limitations. First, we used a self-administered questionnaire which has not been evaluated or validated as a measurement instrument. Discrepancies in how the principal concepts in the survey were presented and understood, respectively, may have led to inaccurate responses. Although the main concepts of the survey were explained in short texts within the questionnaire and the survey was piloted and tested among several participants, the possibility of misunderstandings cannot be excluded.

Second, the recruitment process created a selective study sample. Any generalisation of results must, therefore, be applied with caution. Drawing a convenience sample may have resulted in more people feeling attracted to the survey by the invitation to participate who were interested in the topic. Individuals who are not interested in the topic and their

perspectives may accordingly be underrepresented. Older respondents and those who are low users of the internet, as well as those not standing close to medical associations and societies, might be underrepresented. We do not know how many of the intermediaries, associations or societies that were contacted had passed on the invitation email to members of their network although reminders were sent.

Third, in order to guarantee the anonymity of the participants, no strategy to avoid multiple entries was implemented. However, this may have created bias in the results since it was possible for people to complete the questionnaire multiple times to emphasise their views.

Fourth, respondents were not surveyed about real life situations, but rather rated their general willingness. No specific scenario was presented. Also, no further background information on the baseline situation was provided that would have allowed for an assessment of the necessity ("clinical equipoise") or relevance of a study by the participants. This may have resulted in the weighing of risks and benefits being too abstract, polarised, or not occurring at all, with participants accordingly indicating a high level of willingness. This error raises the possibility of response bias, since it is uncertain whether stakeholders would react similarly if they are asked to engage in a real study.

Fifth, the survey took place during the Covid-19 pandemic. Stakeholders may have been too preoccupied to participate in the survey, or may have considered the topic of the survey to be less relevant to them in the particular situation. Moreover, in a time of urgent need for medical interventions against Covid-19, participants may have been more in favour of benefit assessment post-entry in order to accelerate access to new interventions.

## Conclusion

Patients, patient representatives, healthcare providers and scientists show a high willingness to engage in studies for benefit assessment studies under conditions of simultaneous availability and service provision. Information about the motives to participate in, support or carry out post-entry studies indicates possible starting points for overcoming difficulties and barriers in recruitment, adherence and stakeholder engagement. Additional burden for each stakeholder group, sources of funding, and responsibility are central issues when planning and conducting post-entry studies successfully [22]. Study type was found to be less relevant to stakeholders when considering willingness to engage in a study. However, we observed tendencies that the scientific community and healthcare providers reported a higher willingness to engage in RCTs, but patients and patient representatives reported to prefer OSs.

More research is needed to enhance the knowledge base about post-entry studies in particular, and with regard to study characteristics and implementation strategies. Studies should be conducted in order to evaluate studies for benefit assessments of interventions that are already funded.

## Supporting information

**S1 Appendix. Questionnaire items and original data obtained from survey participants.**
(XLSX)

**S2 Appendix. Contacted associations and societies of stakeholder groups.**
(PDF)

**S3 Appendix. General motives for engagement.** Tables A1-A6 present all general motives mentioned by the respondents of each stakeholder group for or against engagement in post-entry studies.
(PDF)

**S4 Appendix. Study type-specific motives for engagement.** Tables B1-B6 present all study type-specific motives mentioned by the respondents of each stakeholder group for or against engagement in post-entry studies.
(PDF)

**S5 Appendix.**
(XLSX)

## Author Contributions

**Conceptualization:** Julia Stadelmaier, Joerg J. Meerpohl, Ingrid Toews.

**Data curation:** Julia Stadelmaier.

**Formal analysis:** Julia Stadelmaier.

**Funding acquisition:** Joerg J. Meerpohl, Ingrid Toews.

**Investigation:** Julia Stadelmaier, Joerg J. Meerpohl.

**Methodology:** Julia Stadelmaier, Joerg J. Meerpohl, Ingrid Toews.

**Project administration:** Ingrid Toews.

**Software:** Julia Stadelmaier.

**Supervision:** Ingrid Toews.

**Validation:** Julia Stadelmaier.

**Visualization:** Julia Stadelmaier.

**Writing – original draft:** Julia Stadelmaier.

**Writing – review & editing:** Julia Stadelmaier, Joerg J. Meerpohl, Ingrid Toews.

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
