## [Decision Letter · Decision Letter 0]

7 Dec 2021

PONE-D-21-23551Willingness to participate in, support or carry out scientific studies for benefit assessment of available medical interventions: a stakeholder surveyPLOS ONE

Dear Dr. Ingrid Toews

Thank you for submitting your manuscript to PLOS ONE. After careful consideration, we feel that it has merit but does not fully meet PLOS ONE’s publication criteria as it currently stands. Therefore, we invite you to submit a revised version of the manuscript that addresses the points raised during the review process. Kindly go through the reviewer comments carefully and address them as requested and required. Once the revised manuscript is submitted , we might send the article for re-review before finalizing our decision to publish. Thank you for your time and patience.

We look forward to receiving your revised manuscript.

Kind regards,

Paavani Atluri

Academic Editor

PLOS ONE

Journal Requirements:

2. During the internal evaluation of your manuscript we noted that informed consent was not necessary for this study. Please could you clarify whether the IRB specifically waived the need for informed consent for the study.

None

Reviewers' comments:

Reviewer's Responses to Questions

**Comments to the Author**

1. Is the manuscript technically sound, and do the data support the conclusions?

Reviewer #1: Partly

Reviewer #2: Yes

2. Has the statistical analysis been performed appropriately and rigorously? 

Reviewer #1: No

Reviewer #2: Yes

3. Have the authors made all data underlying the findings in their manuscript fully available?

Reviewer #1: Yes

Reviewer #2: Yes

4. Is the manuscript presented in an intelligible fashion and written in standard English?

Reviewer #1: Yes

Reviewer #2: Yes

5. Review Comments to the Author

Reviewer #1: I have read this manuscript with much interest. This is mostly a descriptive study of participants’ willingness to participate, support or carry out scientific studies for benefit assessment of medical interventions. My comments and concerns are listed below.

1- Where exactly was the study conducted? Was there any geographical limitation? Where participants from a specific city, region, country, etc.

2- The introduction is quite short and choppy. It does not lead well to the objectives of the study.

3- The authors devised a 51-item questionnaire for data collection. There is no information on the validity and reliability of the questionnaire. I believe that no psychometric testing been done on the questionnaire.

4- Participants: There is not much information on who the participants are (except a general statement that they are from patients, patient representatives, scientists, healthcare professionals and representatives, and private sector groups). The authors used snow-ball sampling techniques to recruit further participants, but how the primary participants were identified (e.g., from which hospital, university, etc.)?

5- Please revise Table 1 as:

a. Please add row and column totals.

b. I would suggest adding row percentage instead of column percentage.

c. The last block of the table (i.e., Experience) is quite confusing. I would suggest a separate table for this section with proper title.

Reviewer #2: The authors conducted an observational study to investigate the willingness of relevant stakeholder groups to participate or engage in post-entry studies. There are, at any rate, several issues to be addressed as follows.

1) “New, innovative interventions…”, there is an extra comma.

2) Sampling was not described in sufficient details. have the authors applied inclusion and exclusion criteria? How did the author select the target organisations?

3) “The completion rate was 161 calculated using the ratio of people who had submitted the final questionnaire page and those”, have the authors compared the two groups?

4) “For comparison between two groups an independent t-test was 167 used for a response frequency n > 30, and the Mann-Whitney-U-Test for n < 30 162”, this reviewer understands that Excel was used for descriptive analysis but what statistical tool was used for statistical tests?

5) Have the authors checked the effect of age and gender on responses?

6) For reasons that might or might not be under the authors' control, the figures have a low-resolution appearance

6. PLOS authors have the option to publish the peer review history of their article (what does this mean?). If published, this will include your full peer review and any attached files.

---

## [Author Response · Author response to Decision Letter 0]

21 Jan 2022

We thank you for your advice. We revised our manuscript accordingly.

We rechecked our manuscript against the requirements of the journal and revised it where needed.

2. During the internal evaluation of your manuscript we noted that informed consent was not necessary for this study. Please could you clarify whether the IRB specifically waived the need for informed consent for the study. 

Response: Thanks for highlighting this issue. In its written statement of reasons for the vote, the ethics committee of the Albert-Ludwigs-University pointed out that it was assumed that no personal data but only anonymised data would be collected and evaluated. Accordingly, the need for informed consent was waived. We address this issue by complementing our methods section with the following information: 

“No personal but only anonymised data were collected and analysed for the purpose of this study. Therefore, no informed consent for the study was necessary according to the standards of the Ethics Committee of the Albert-Ludwigs-University Freiburg and the data protection officer.” (ll.136-139)

None

Response: We thank you for the changes you make on our behalf with regard to the statement about competing interests. We state that the authors have declared that no competing interests exist and will make sure that this information is also included where needed during the resubmission

Reviewers comments Reviewer 1: 

I have read this manuscript with much interest. This is mostly a descriptive study of participants’ willingness to participate, support or carry out scientific studies for benefit assessment of medical interventions. My comments and concerns are listed below. 

Response: Thank you for your feedback and your support in improving this manuscript.

1- Where exactly was the study conducted? Was there any geographical limitation? Where participants from a specific city, region, country, etc. 

Response: Thank you for pointing out these questions. We did no limit the survey to a specific geographical area. However, since the survey was in German language and recruiting was conducted mainly through German associations and societies, we assume that respondents are German-speaking people in Germany. We added this information in the method section (subheading ‘sampling’) as follows:

“Members of the following stakeholder groups were the target population of the survey: (i) patients or patient representatives, (ii) healthcare providers (physicians and health professionals), (iii) scientists and trialists in medical and health science, and (iv) representatives of the medical or pharmaceutical private sector (e.g. medical technology or the pharmaceutical industry). No geographical limitation was applied. However, as the questionnaire was written in German, knowledge of the German language was necessary to participate in the survey.” (ll.148-154)

2- The introduction is quite short and choppy. It does not lead well to the objectives of the study.

Response: To improve this section, we have added some informative sentences and moved the second paragraph down. This resulted in a number of edits that can be seen in the accompanying revised manuscript. If you still have concerns with regard to specific aspects within the introduction, we would appreciate more details about suggested changes.

3- The authors devised a 51-item questionnaire for data collection. There is no information on the validity and reliability of the questionnaire. I believe that no psychometric testing been done on the questionnaire. 

Response: Thank you for your comment. In order to provide more information on the questionnaire we added the following sentences in the method section (subheading ‘questionnaire’): 

“In the course of the systematic literature searches we did not identify any suitable, validated questionnaires for the specific purpose of our study. Therefore, we designed a de-novo self-administrated questionnaire in German language.” (ll.104-106) 

“We did not perform psychometric testing to validate our questionnaire.” (l.122)

4- Participants: There is not much information on who the participants are (except a general statement that they are from patients, patient representatives, scientists, healthcare professionals and representatives, and private sector groups). The authors used snow-ball sampling techniques to recruit further participants, but how the primary participants were identified (e.g., from which hospital, university, etc.)? 

Response: We agree with you and we added more information about the sampling and associations we contacted. Moreover, we attached a list with details of the contacted associations and societies to the S2 Appendix.

“A google search was performed to identify relevant associations and societies representing the target stakeholder groups in Germany. In order to recruit patients and patient representatives, the office of patient representatives in the German Federal Joint Committee (Gemeinsamer Bundesausschuss, G-BA), as well as the four organisations that are currently entitled to appoint patient representatives to the G-BA were contacted. Healthcare providers (physicians, dentists, psychologists, nurses, allied health professionals) were contacted via associations representing the interests of their respective profession. Scientists and trialists were recruited via representatives of the German network for evidence-based medicine, the German network for healthcare research and the Clinical Trials Unit of the University of Freiburg. Trade organisation for the medical technology and pharmaceutical industries were approached in order to invite representatives of the private sector. The contacted associations and societies are listed in detail in the S2 Appendix.” (ll.157-168)

5- Please revise Table 1 as:

a. Please add row and column totals.

b. I would suggest adding row percentage instead of column percentage.

c. The last block of the table (i.e., Experience) is quite confusing. I would suggest a separate table for this section with proper title. 

Response: Thank you for this thoughtful comment. To complement this table, we have added row and column totals and moved information about experience to new created tables (Table 2A and 2B). 

We did not include row percentages since several respondents identified to belong to more than one stakeholder group, the sum of row percentage does no attain 100 per cent. We added an explanation of this issue as footnote to our table.

Reviewer 2: 

The authors conducted an observational study to investigate the willingness of relevant stakeholder groups to participate or engage in post-entry studies. There are, at any rate, several issues to be addressed as follows. 

Response: Thank you for your feedback and your support in improving this manuscript. 

1) “New, innovative interventions…”, there is an extra comma. 

Response: Thanks. We made this edit.

2) Sampling was not described in sufficient details. have the authors applied inclusion and exclusion criteria? How did the author select the target organisations? 

Response: We agree with you. As described above, we added more information about the sampling and the associations we contacted (ll.148-154; ll.157-168).

3) “The completion rate was 161 calculated using the ratio of people who had submitted the final questionnaire page and those”, have the authors compared the two groups? 

Response: We did not compare the responses of those who submitted the final questionnaire page and the non-completers. First, the completers include seven respondents that were screened out. Second, among the 46 respondents who did not submit the final page, 24 were excluded from our analysis since they did not fulfil the inclusion criteria, that is, they responded to one question only. With 22 respondents remaining (distributed among four stakeholder groups), we observe that this sample size of this subgroup is too small to conduct further post-hoc analyses to obtain meaningful results.

4) “For comparison between two groups an independent t-test was 167 used for a response frequency n > 30, and the Mann-Whitney-U-Test for n < 30 162”, this reviewer understands that Excel was used for descriptive analysis but what statistical tool was used for statistical tests? 

Response: Thank you for this comment. We used Microsoft Excel for both, descriptive and analytical statistics. We rephrased the sentence in our method section as follows: “All valid responses were tabulated, and statistical analyses were conducted using Microsoft Excel 2010.” (ll.185-186)

5) Have the authors checked the effect of age and gender on responses?

Response: Thank you for raising this question. We did not carry out subgroup analyses due to the small size of the sample. We had data from a total of 154 respondents (distributed among different stakeholder groups). From a methodological point of view, we think that this sample size is not large enough for further analyses. We addressed this issue in our result section as follows:

“We did not carry out further subgroup analyses on the effect of sociodemographic variables since the number of respondents in each stakeholder group is too small to create further representative subgroups and perform reasonable analyses.” (ll.272-274)

6) For reasons that might or might not be under the authors' control, the figures have a low-resolution appearance

Response: Thank you for highlighting this issue. The figure was edited according to the journal’s requirements. In case there are inadequacies, we are happy to make changes to our figure.

---

## [Decision Letter · Decision Letter 1]

15 Jun 2022

PONE-D-21-23551R1Willingness to participate in, support or carry out scientific studies for benefit assessment of available medical interventions: A stakeholder surveyPLOS ONE

Dear Dr. Toews,

Thank you for submitting your manuscript to PLOS ONE. After careful consideration, we feel that it has merit but does not fully meet PLOS ONE’s publication criteria as it currently stands. Therefore, we invite you to submit a revised version of the manuscript that addresses the points raised during the review process.

We look forward to receiving your revised manuscript.

Kind regards,

Paavani Atluri

Academic Editor

PLOS ONE

Journal Requirements:

Reviewers' comments:

Reviewer's Responses to Questions

**Comments to the Author**

1. If the authors have adequately addressed your comments raised in a previous round of review and you feel that this manuscript is now acceptable for publication, you may indicate that here to bypass the “Comments to the Author” section, enter your conflict of interest statement in the “Confidential to Editor” section, and submit your "Accept" recommendation.

Reviewer #1: All comments have been addressed

Reviewer #3: All comments have been addressed

2. Is the manuscript technically sound, and do the data support the conclusions?

Reviewer #1: Yes

Reviewer #3: Partly

3. Has the statistical analysis been performed appropriately and rigorously? 

Reviewer #1: Yes

Reviewer #3: Yes

4. Have the authors made all data underlying the findings in their manuscript fully available?

Reviewer #1: Yes

Reviewer #3: Yes

5. Is the manuscript presented in an intelligible fashion and written in standard English?

Reviewer #1: Yes

Reviewer #3: Yes

6. Review Comments to the Author

Reviewer #1: I have reviewed the revision. The authors have addressed all of my comments. I have no further comment.

Reviewer #3: I wasn't one of the original reviewers, but as far I as understand, the authors have addressed the points raised by the reviewers.

I have some minor observations having read the revised paper.

1. Abstract and keywords: from the keywords and abstract, I'm not sure I would pick up that this paper has its roots in HTA, decision-making and MEA's, which is explained clearly in the Introduction. I might miss it in a title and abstract screen for a systematic review of barriers to MEAs, for example. The authors might consider some adjustments to broaden their impact; I leave that to their judgement.

2. Table 2: I found this a bit confusing - were the stakeholders asked about their experience in different units? It might help to explain in the legend.

3. I wasn't sure how the evidence supported that patients preferred observational studies or that healthcare providers were inexperienced - might help to point it out earlier than the Discussion or Conclusion.

4. I think there's an interesting paradox in the discussion, between willingness to participate in a study on an available treatment (because they assume it's therefore OK) and preference to avoid getting placebo when there's a known treatment available (the equipoise issue). The authors could make a stronger case for (their) future work on this if they chose to; again I leave that to their judgement.

7. PLOS authors have the option to publish the peer review history of their article (what does this mean?). If published, this will include your full peer review and any attached files.

Reviewer #1: No

Reviewer #3: No

---

## [Author Response · Author response to Decision Letter 1]

29 Jun 2022

We appreciate the valuable comments made by the editor and peer reviewers and their support in improving this manuscript. We have addressed their comments in full and revised our manuscript accordingly.

Editors comment. 

Response: We thank you for your advice. We rechecked our reference list and revised it where needed. We added the corrections of the articles of Sheridan et al. 2020 and Eysenbach 2012 to the reference list:

Sheridan R, Martin-Kerry J, Hudson J, Parker A, Bower P, Knapp P. Why do patients take part in research? An overview of systematic reviews of psychosocial barriers and facilitators. Trials. 2020;21(1):840. doi: 10.1186/s13063-020-04793-2. Corrected and republished from: Trials. 2020;21(1):259. doi: 10.1186/s13063-020-4197-3.

Eysenbach G. Improving the quality of Web surveys: the Checklist for Reporting Results of Internet E-Surveys (CHERRIES). J Med Internet Res. 2012;14(1):e8. doi: 10.2196/jmir.2042. Corrected and republished from: J Med Internet Res. 2004;6(3):e34. doi: 10.2196/jmir.6.3.e34.

Reviewer #3: 

I have some minor observations having read the revised paper.

Response: Thank you for your feedback and your support in improving this manuscript. 

1. Abstract and keywords: from the keywords and abstract, I'm not sure I would pick up that this paper has its roots in HTA, decision-making and MEA's, which is explained clearly in the Introduction. I might miss it in a title and abstract screen for a systematic review of barriers to MEAs, for example. The authors might consider some adjustments to broaden their impact; I leave that to their judgement.

Response: Thank you for this comment. We added the following phrases to the abstract: 

“Post-entry studies are a key element in managed entry agreements and aim at generating evidence about the additional benefit of new medical interventions before reimbursement decisions are made.” (ll.20-22)

Moreover, we added “managed entry agreements” to the list of keywords.

2. Table 2: I found this a bit confusing - were the stakeholders asked about their experience in different units? It might help to explain in the legend.

Response: Thank you for raising this question. The item ‘experience’ differed between stakeholder groups: We asked patients, healthcare providers and scientists about the number of participation/ engagements in studies of medical research, whereas for patient representatives and representatives of the private sector we asked for their experience in medical research in years. We added this information to the methods section as follows: 

“(iv) demographic and professional background including experience in years (for patient representatives and representatives of the private sector) or number of engagements in studies of medical research (for patients, healthcare providers and scientists)”(ll.119-122)

Furthermore we redefined the title and added a legend to Table 2A and 2B:

Table 2A. Engagement of the respondents in medical research (in number of studies)a

Table 2B. Experience of the respondents in medical research (in years)a

a Wording and scales of the item ‘experience’ differed between the respective stakeholder groups. (ll. 224-227)

3. I wasn't sure how the evidence supported that patients preferred observational studies or that healthcare providers were inexperienced - might help to point it out earlier than the Discussion or Conclusion.

Response: Thank you for highlighting this issue. Information about the characteristics of the respondents is displayed in Table 1, 2A and B. We agree with you in highlighting some results regarding the characteristics in the text, and added the following phrases:

“The majority of respondents were female (52.6%), aged between 35 and 54 years (46.8%), and reported to have little experience with regard to participation or engagement in medical research.” (ll. 228-230)

Differences in the willingness to participate or engage in RCTs or observational studies are already described in the results section ‘Willingness with regard to study type’ (ll.251ff). Since significant differences were only observed in the group of the scientists, we toned down our wording in the conclusion section to the following:

“Study type was found to be less relevant to stakeholders when considering willingness to engage in a study. However, we observed tendencies that the scientific community and healthcare providers reported a higher willingness to engage in RCTs, but patients and patient representatives reported to prefer OSs.” (ll. 413-416)

4. I think there's an interesting paradox in the discussion, between willingness to participate in a study on an available treatment (because they assume it's therefore OK) and preference to avoid getting placebo when there's a known treatment available (the equipoise issue). The authors could make a stronger case for (their) future work on this if they chose to; again I leave that to their judgement.

Response: Thank you for this comment. We added the following: 

“Future research that specifies details of the comparator (active or inactive) might therefore be indicated. This could contribute to resolving the paradox between high willingness of study participation as reported in our survey and other research findings that demonstrate a reluctance of study participation in general.” (ll. 354-357)

---

## [Editor Report · Decision Letter 2]

8 Jul 2022

Willingness to participate in, support or carry out scientific studies for benefit assessment of available medical interventions: A stakeholder survey

PONE-D-21-23551R2

Dear Dr. Toews,

We’re pleased to inform you that your manuscript has been judged scientifically suitable for publication and will be formally accepted for publication once it meets all outstanding technical requirements.

Kind regards,

Paavani Atluri

Academic Editor

PLOS ONE
---

## [Editor Report · Acceptance letter]

3 Aug 2022

PONE-D-21-23551R2 

Willingness to participate in, support or carry out scientific studies for benefit assessment of available medical interventions: A stakeholder survey 

Dear Dr. Toews:

I'm pleased to inform you that your manuscript has been deemed suitable for publication in PLOS ONE. Congratulations! Your manuscript is now with our production department. 

Kind regards, 

on behalf of

Dr. Paavani Atluri 

Academic Editor

PLOS ONE